# Association between Educational Level and Physical Activity in Chronic Disease Patients of Eastern Slovakia

**DOI:** 10.3390/healthcare9111447

**Published:** 2021-10-26

**Authors:** Alena Buková, Erika Chovanová, Zuzana Küchelová, Jan Junger, Agata Horbacz, Mária Majherová, Silvia Duranková

**Affiliations:** 1Institute of Physical Education and Sport, Pavol Jozef Šafárik University, 040 11 Košice, Slovakia; zuzana.kuchelova@upjs.sk (Z.K.); jan.junger@upjs.sk (J.J.); agata.horbacz@upjs.sk (A.H.); 2Faculty of Sport, University of Prešov, 080 01 Prešov, Slovakia; erika.chovanova@unipo.sk; 3Faculty of Humanities and Natural Sciences, University of Prešov, 080 01 Prešov, Slovakia; maria.majherova@unipo.sk (M.M.); silvia.durankova@unipo.sk (S.D.)

**Keywords:** educational level, chronic disease patients, leisure time activity, patients’ awareness, physical activity, prevention

## Abstract

Aims: This study aimed to investigate selected chronic diseases patients with different educational attainment regarding their awareness of and compliance with recommended physical activity. Method: This cross-sectional study was conducted from October 2018 to February 2019 in cooperation with chronic diseases clinics in eastern Slovakia. The study involved 893 patients. Results: People with higher education apparently recognise to a greater extent the importance of nutrition, diet, and the role of physical activity in treating their disease. Moreover, they have knowledge of physical exercises appropriate for their disease. Conversely, a noticeably higher number of less educated patients reported receiving general, respectively detailed information about the importance of physical activity in treating their disease. Differences in awareness of appropriate exercises and their implementation were not statistically significant. Conclusions: The results fail to prove educational attainment being a key determinant of chronic diseases. However, it can be reasonably argued that lower educational attainment may be a reliable risk signal of chronic diseases in later life.

## 1. Introduction

Educational attainment is directly related to the socio-economic status of the individual, as well as that of the country in which he or she lives. Low income and unstable job statuses, which are likely to be linked to low educational level, are reasonably predicted to raise the risk of chronic diseases (NCD) incidence [1]. According to Beltrán-Sánchez and Andrade [2], these diseases are associated with a large financial burden on the national economy of countries owing to increases in health care costs and health care utilisation that threaten the stability of the public health system. NCDs such as hypertension, coronary heart disease, circulatory diseases, and cancer are the most common causes of death not only in the European Union, but also in Slovakia [3,4,5]. Adherence rates are influenced by several factors, including patients’ lack of understanding of their disease, which is closely related to the patients’ educational level [6,7]. The inverse association between education and NCDs has been confirmed by several studies. This relates to cardiovascular diseases [8,9,10,11]; type 2 diabetes [12,13,14]; and some forms of cancer, especially gastric cancer [15] and chronic lung disease [16,17], as well as to stroke [1], kidneys [18], and liver [19,20].

Patients’ adherence to treatment recommendations prescribed by their physician has positive and discernable effects on treatment outcomes [21,22]. According to several authors, higher education is correlated with lower risk of chronic diseases or even causes of death as a result of obtaining more information about health risks, preventive measures, and access to treatment methods [7,19,23,24]. However, as Kooi et al. [25] found in a health survey sample of 49 countries, health inequalities attributable to differences in education are greater in more developed countries.

Physical activity (PA) is a critically underrated disease-prevention strategy that has widespread health benefits, not only in the prevention of diseases in healthy people [26,27,28,29], but also in patients suffering from various NCD chronic diseases [30,31,32,33,34,35]. At the same time, it is obvious that exercise is a powerful medicine for the primary and secondary prevention of virtually every NCD and for reducing premature mortality [36]. The World Health Organization has identified increasing PA as one of four key strategies for reducing the global NCD epidemic. On the other hand, inactivity and sedentary lifestyles have well-known detrimental effects on all-cause mortality and chronic-disease morbidity in adults [36,37].

Taking into account the knowledge gained and the limitations of the existing literature, the present study conducted an investigation into patients’ awareness and fulfilment of recommendations for performing PA by selected NCD patients in relation to their educational attainment.

## 2. Materials and Methods

### 2.1. Study Population

The survey was conducted from 10/2018 to 2/2019 at outpatient clinics in eastern Slovakia and comprised patients from 19 cardiology, 14 metabolism, and 9 oncology clinics. We obtained written permission from the representatives of all the clinics in advance to contact and approach patients. At baseline, a trained administrator explained the aim and procedure of the research, and patients willing to participate in the research signed a written consent to participate in the study. They then completed a questionnaire regarding their health status and PA. After completing the questionnaire, the patients were informed about the benefits of PA in relation to their disease and instructed on the most appropriate PA for their disease. Subsequently, the patients were given a leaflet with general instructions about PA according to the FITT principles (frequency, intensity, time, type) [38]. We randomly addressed 1193 adult patients treated in these clinics, of whom 282 refused to participate in the survey. We further excluded another 18 patients for not meeting one or more of the essential criteria listed below. The criteria for participating in the research were met by 893 patients—353 males and 540 females. Among all participants, 8.29% patients acknowledged more than one chronic disease. The average age of respondents was 54.24 years, with the youngest patient being 18 years old and the oldest 90 years old. Other socio-demographic indicators are presented in Table 1.

#### 2.1.1. Exclusion and Inclusion Criteria

Inclusion criteria:

Patients were enrolled in the research after having met the criteria below:-Over 18 years of age;-The occurrence of one or more diagnoses of three underlying diseases of affluence that do not prevent physical activity (cardiovascular disease, oncological disease, metabolic disease);-Diagnosis having been treated by a specialist for a minimum of 1 year;-Willingness to give informed consent to participate in the research;-Willingness to fill in questionnaires regarding physical activity and be provided information about physical activity for a given diagnosis.

Chronic disease included the following:MD—metabolic diseases—diabetes or high blood sugar, obesity, thyroid disorders;CVD—cardiovascular diseases—heart attack, including myocardial infarction, coronary thrombosis, and any other heart problems including congestive heart failure;OD—oncological diseases—cancer of any subtype—cancer or malignant tumor, including leukemia or lymphoma.

Exclusion criteria:-Bed-bound patient status;-Diagnoses for which PA is not recommended;-Malignancy (active cancer) or life-threatening disease;-Participation or plan to participate in another study.

#### 2.1.2. Sample Size

In the year under review, 851,623 patients were registered in the outpatient clinics of eastern Slovakia and were treated for at least one of the monitored chronic diseases. We calculated the minimum sample size according to the estimation given in Daniel [39], where *n* = Z2P(1 − *p*)/d2 (Z = 2.576 for 99% level of confidence; *p* = 0.5 for expected sample proportion of 50%; d = 0.05 for the 5% margin of error). Based on this calculation, the minimum number was set at *n* = 667 patients. We set a higher number of *n* = 990 as we anticipated a 30–50% loss. Thereof, our sample of 893 respondents is representative considering the number of patients at outpatient clinics in eastern Slovakia.

Table 2 specifies the survey sampling method: the number of outpatient clinics in both regions of eastern Slovakia in the observed year, the total number of clinics contacted, and clinics that gave written consent. It also presents the number of patients treated for the diseases we monitored, the number of patients who gave written consent, and the number of patients who refused to participate in our survey and of those excluded owing to not meeting the above criteria.

### 2.2. Sample and Procedure

All relevant data were collected using a non-standardized questionnaire, part of a questionnaire battery explicitly designed for this research. The questionnaire was partly based on validated instruments from other, mostly physical activity studies (e.g., PA questions from the GPAQ [40]). The questionnaire contained 29 questions and was designed for a total completion time of 20 min. For the purposes of this questionnaire, we selected questions regarding patients’ awareness of PA and its performance:-What do you think is most important in the prevention and treatment of your health problem? (you can tick more than one option)-Can regular physical activity have a positive effect on your health problem?-Have you ever been told by your doctor or nursing staff about the importance of physical activity in the prevention and treatment of your health problem?-Do you have information that there exist minimum requirements for regular physical activity even for at-risk patients?-Do you know the appropriate physical exercises for the prevention and treatment of your health problem?-What do you do most in your leisure time? (you can tick more than one option)-In a typical week, on how many days do you do moderate-intensity sports, fitness or recreational (leisure) activities? “Typical week” means a week when the participant is engaged in his/her usual activities.-How much time do you spend doing moderate-intensity sports, fitness or recreational (leisure) activities on a typical day?-Do you do any moderate-intensity sports, fitness or recreational (leisure) activities that cause a small increase in breathing or heart rate such as brisk walking, [cycling, swimming, volleyball] for at least 10 min continuously? Name 1 preferred activity.

The majority of questions included were closed-ended, with the option for respondents to elaborate on a certain response, and were of a factual nature. The first 7 questions focused on socio-demographic indicators, 4 questions covered patients’ medical condition and awareness of their medical condition, 14 questions dealt with PA and awareness of PA, and 4 questions focused on selected lifestyle factors.

Processing of statistical data was performed using IBM SPSS version 23 (Reference: IBM Corp. Released 2015. IBM SPSS Statistics for Windows, Version 23.0. Armonk, NY, USA). Pearson’s Chi square test was used to determine differences in the actual frequency of the occurrence of attributes. Given the low number of patients with primary and vocational education (14.4% in total), and having considered the high predisposition of occupational relatedness in relation to manual work, we merged these two categories when calculating the statistical significance. The Chi square test (*χ*^2^) was used to test the agreement of frequencies in the occurrence of each variable in relation to education. For questions regarding leisure time activity and duration of PA, we used the nonparametric Kruskal Wallis test. Testing of the statistical hypothesis was performed at the significance level of α < 0.05. Cramer’s V was calculated as a measure of effect size. It takes values in the range of 0–1. Estimation of the effect size was done according to the following criteria (with df = 2): =0.07 represents a small effect, =0.21 represents a medium effect, and =0.35 represents a large effect.

The research conducted in 2017–2019 was supported by grant project No. 1/0825/17 ‘Recommendations for physical activities in prevention and control of non-communicable diseases and their implementation in the eastern part of Slovakia.’ and grant project No 1/0120/19 ‘Movement correction of problem behavior of pupils of the standard population and pupils with special educational needs educated in conditions of integration’.

The protocol was approved by the Human Research Ethics Committee of Pavol Jozef Šafárik University in Košice [approval No. PJSU-0825/17-1].

## 3. Results

Table 3 presents the subjective views of patients with selected chronic diseases on PA in relation to their level of education. We found that education was evidently related to patients’ views on the importance of PA (χ^2^ = 39.8118, sv = 8; *p* < 0.01; V = 0.1493—medium effect). Although the majority of respondents reported that nutrition and diet were of greatest importance, a higher percentage of those with a university education also attributed more value to PA. Similarly, education was related to respondents’ subjective opinion of whether PA affects their health status (χ^2^ = 48.9281, sv = 8; *p* < 0.01; V = 0.1655—medium effect). The majority of respondents, predominantly with a university education, reported that it unequivocally does. However, a considerably high percentage of respondents either could not assess this question or did not give it a second thought. We inquired whether patients received information about the importance of PA in the treatment of their disease. Most patients reported receiving general or even detailed information. Interestingly, according to patients’ statements, detailed information is mainly provided to patients with less education. Based on mathematical and statistical examination, we can confirm a statistically significant relationship between the above question and patients’ education (χ^2^ = 24.0185, sv = 10; *p* < 0.01; V = 0.1159—medium effect). On the contrary, a significantly higher number of university-educated patients, according to their subjective statement, are familiar with appropriate physical exercises in relation to their disease (χ^2^ = 48.9281, sv = 8; *p* < 0.01; V = 0.1655—medium effect). No differences were found in relation to education when asked about awareness of minimum PA requirements (χ^2^ = 14.5845, sv = 8; *p* = 0.06; V = 0.0903—medium effect).

With regard to the health of patients, the way they spend their leisure time is of great importance. Thus, our concern in the next question was how patients with chronic disease spend their leisure time and whether there was a difference in leisure time in relation to patients’ education. The vast majority of patients either do household chores or spend time passively watching TV or at the PC (Table 4). Despite these preferences, we found statistically significant differences between leisure time and educational level (χ^2^ = 32.5096, sv = 10; *p* < 0.01; V = 0.1351—medium effect). In addition to the two most preferred activities in all social groups, patients with primary education were more likely to shop and listen to music; university-educated patients were more likely to read books, but also to engage in recreational PA.

Many authors confirm the manifest relationship between active leisure time, inclusive of regular PA, and human health [27,28,29,30,31,32,33,34,35,36]. Thus, in the last set of questions, we were interested in the frequency, duration, and mainly the type of PA performed. We aimed to find a difference in the association between education and the implementation of PA in the week (Table 5). The observed patients perform PA mostly irregularly, with lower education being dominant. When merging the frequencies of performing PA at least once a week and more, we observe a difference in activity in patients with primary and vocational education (45.2%), and secondary education (55.2%), to the advantage of patients with a completed higher education (67.4%). Despite the differences between the groups in all variables, these were not statistically significant, whether for the frequency of PA (Kruskal–Wallis = 5.9167, *p* = 0.0519; η^2^ = 0.0044—small effect), for the duration of PA (Kruskal–Wallis = 0.6430953; *p* = 0.7250; η^2^ = 0.0015—small effect), or for the type of PA performed PA (χ^2^ = 14.2515, sv = 12; *p* = 0.2849; V = 0.0894—medium effect).

## 4. Discussion

The present study aimed to investigate selected chronic disease patients with different educational attainment regarding their awareness of and compliance with recommended physical activity. We agree with the assertion of Cutler et al. [23], who state that there is a persistent and substantial association between education and health. While most of the literature focuses on individual lifestyle factors in relation to educational level, we have not found studies that surveyed patients’ awareness of the importance of preventing and treating their disease through PA, and awareness of their appropriate PA options, as related to their level of education. Consistent with our results, we can confirm that people with a higher education are evidently more likely to recognize the importance of nutrition and diet, as well as the importance of PA in the prevention and treatment of their disease. The majority of patients, regardless of education, reported receiving general or even detailed information about the importance of PA in the treatment of their disease, with a noticeably higher number of patients with lower education. On the contrary, a demonstrably higher number of university-educated patients, according to their subjective assertion, are aware of appropriate physical exercises relevant to their disease. We have not found statistically significant differences between patients’ education in terms of awareness of appropriate physical exercises, nor in the implementation of PA (its frequency, duration, and the type of PA performed).

Motivating people to make behavioral changes that would bring health benefits is difficult [41]. According to the above authors, a large number of individuals are not motivated enough to engage in health-promoting behaviors. Currently, the AHA (American Heart Association) recommends that at-risk individuals maintain a healthy diet that encourages intake of vegetables, fruits, legumes, nuts, whole grains, and fish to optimize CVD risk factor profiles [42]. In this regard, we can state that the majority of probands involved in the study, according to their statements, follow the recommendations in relation to nutrition and placed great emphasis on nutrition and diet in the treatment of their disease. This is predominantly true of the group of university-educated patients. Close to one-third of the participants involved in the study attributed importance to regular sleep and rest in the treatment of their disease, which corresponds with the findings of Wang et al. [43]. Based on their research, the aforementioned authors support the importance of sleep in the modification of lifestyle and health in CVD patients. According to them, 6–8 h of nightly sleep is associated with the lowest risk of death and major CVD prevalence.

People with chronic diseases should predominantly perform aerobic activities without oxygen debt, such as brisk walking, swimming, cycling, skating, and so on, as such activities primarily involve large muscle groups that discernibly affect the circulatory and respiratory systems. These activities are predominantly undertaken by patients with a university education (over 90% of them, compared with 60% of those with primary and vocational education). Domestic chores are, according to Frontera [44], the most frequently performed physical activity in general, not only in the elderly or in people suffering from some form of civilisation disease. This fact corresponds with our results. The most frequent leisure activities in all groups, apart from household chores, were watching TV and using the PC, i.e., passive leisure activities.

Every physician involved in the treatment and prevention of CVD should, according to Sallis et al. [36], commit to making his or her patients more active. In our assertion, this applies to all physicians who treat patients with NCDs. Obviously, it only applies if the patient’s condition allows so and PA is not a contraindication in his/her treatment. The treatment of a patient with any of the NCDs should include not only dietary counselling, but also counselling in relation to the implementation of PA. Building on existing evidence, it is clear that regular PA should be considered as first-line treatment, used for both the treatment and prevention of chronic diseases.

Previous research [45] suggests that activity adequacy mindsets may influence physical activity behavior and health. However, in most cases, experts indicate time as a major barrier to the delivery of preventive services [46,47,48]. It has also been postulated that a lack of knowledge and training among practitioners, as well as clinical inertia, may lead to incomplete adherence to practice guidelines [47,49,50].

Although some studies have reported higher physical activity in university-educated people in various domains [51,52,53], in our study of NCD patients, where PA is expected to constitute an important and integral part of their treatment, we did not confirm that finding. Although the patients with higher education performed aerobic PA more often, the difference compared with the other groups was not statistically significant. Although the patients involved in our research did not place great importance on regular PA, it is noteworthy that, when asked directly about the impact of PA on their health status, they acknowledged the importance of this lifestyle factor. Such a response corresponds with the finding of Morrow et al. [54], who suggest that knowledge of the recommendations does not predict increased engagement in physical activity.

According to Wen et al. [55], a better understanding of mindsets may lead to formulating more effective physical activity recommendations. For instance, recommendations could encourage individuals to meet optimal amounts of activity, while affirming that they can gain substantial health benefits even at lower levels of activity. Recommendations that encourage an adaptive mindset in addition to behavior change are more likely to foster healthy lifestyles and well-being. Recommendations are also a valuable tool to encourage people to reach health-promoting activity levels. Nevertheless, we agree with Lindheimer et al. [56], who suggest that the mere belief that one engages in exercise accounts for half of the mental health benefits of exercise, which is considered just as important as physical health.

## 5. Conclusions

Overall, the above findings underline the importance of education level as a sound and clear predictor of NCD prevalence. Despite the fact that we did not find a positive correlation between the educational level of chronic patients and the actual implementation of PA (namely, its frequency, duration, and the type of PA performed), we can still argue, drawing also on other research, that exercise is not only excellent an means of prevention against various NCDs, but also has an impact on the course of the disease and substantially promotes the positive effects of the treatment.

### Strengths and Limitations of the Study

The strength of the study is the sample size, which is representative considering the number of patients at outpatient clinics in eastern Slovakia.

According to our findings, this study is one of the very few, if not the only one, in Slovakia and Europe ascertaining patients’ awareness of the importance of PA in both the prevention and treatment of their disease, awareness of appropriate exercise in relation to their disease, while examining leisure time activity with regard to patients’ education, as well as the actual implementation of proper PA with respect to the educational level of patients with particular NCDs.

Limitations include the ‘healthy cohort’ effect, which may mean the findings are less generalisable to the general population. Our study was based on data from self-reported questionnaires, which are unambiguously less objective than findings obtained from clinical interview, physical examination, or advanced imaging. A further, major limitation of the study is the choice of questions in the questionnaire, which has not been validated. The questions were adopted from other validated questionnaires.

## Figures and Tables

**Table 1 healthcare-09-01447-t001:** Percentage distribution of patients by demographic characteristics and gender (*n* = 893).

Characteristics	Male (%)	Female (%)	Both Sexes (%)
Number of respondents	353 (38.6)	540 (61.4)	893 (100)
Age group (years)			
<30	63 (17.8)	47 (8.7)	110 (12.3)
30–39	46 (13)	51 (9.4)	97 (10.9)
40–49	65 (19)	94 (17.4)	161 (18)
50–59	58 (16.4)	83 (15.4)	141 (15.8)
60–69	64 (18.1)	134 (24.8)	198 (22.2)
70–79	43 (12.2)	102 (18.9)	145 (16.2)
>80	12 (3.4)	29 (5.4)	41 (4.6)
Disease			
metabolic	166 (47)	241 (44.6)	407 (45.6)
cardiovascular	161 (45.6)	223 (41.3)	384 (43)
oncological	26 (7.4)	76 (14.1)	102 (11.4)
Duration of health problem			
0–5 years	120 (34)	160 (29.6)	280 (32.9)
6–10 years	121 (34.3)	175 (32.4)	296 (31.2)
11–15 years	70 (19.8)	98 (18.1)	168 (18.9)
16–20 years	26 (7.4)	62 (11.5)	88 (10.0)
21 and more years	16 (4.5)	45 (8.3)	61 (6.8)
Education			
elementary	15 (4.2)	14 (2.6)	29 (3.2)
vocational	42 (11.9)	57 (10.6)	99 (11.2)
HS graduate	179 (50.7)	292 (54.1)	471 (52.7)
university	117 (33.1)	177 (32.8)	294 (32.9)
Occupation			
permanent	174 (49.3)	214(39.6)	388 (43.1)
occasional	19 (5.4)	3 (0.6)	22 (2.5)
unemployed	12 (3.4)	24 (4.4)	36 (4)
student	35 (9.9)	16 (3)	51 (5.7)
retired	113 (32)	283 (52.4)	396 (44.3)
Occupation			
sedentary	134 (38)	170 (31.5)	304 (34)
physically demanding	68 (19.3)	70 (13)	138 (15.5)
other	19 (5.4)	18 (3.3)	37 (4.1)
none	132 (37.4)	282 (52.2)	414 (46.4)
Residence			
urban	239 (67.7)	348 (64.4)	587 (65.7)
rural	114 (32.3)	192 (35.6)	306 (34.3)

**Table 2 healthcare-09-01447-t002:** Survey sampling method.

Clinics	Diabetes and Endocrinology Clinics			Cardiology Clinics			Oncology Clinics		
	KE *	PO *	Total	KE *	PO *	Total	KE *	PO *	Total
Number of clinics *	51	39	90	58	38	96	15	16	31
Approached clinics	11	9	20	19	7	26	9	4	13
Clinics—consent	8	6	14	16	3	19	7	2	9
Number of patients/thousand	189.82	167.11	356.94	217.81	136.03	353.84	**	**	140.85
Patients—consent	225	182	407	336	48	384	89	13	102
Patients—refusal	53	28	91	83	24	107	52	32	84
Criteria not met	4	0	4	8	6	14	0	0	0

* including children’s clinics. KE *—Košice region; PO *—Prešov region. ** available are only joint data from Slovakia (both regions merged).

**Table 3 healthcare-09-01447-t003:** Opinions of patients with selected NCDs in relation to education.

	Educational Level %				
Patients’ Opinions	Elementary (*n* = 29)	Vocational (*n* = 99)	HS Graduate (*n* = 471)	University (*n* = 294)	Total (*n* = 893)
Importance in prevention and treatment					
nutrition, eating	51.7	39.4	65.6	70.7	56.9
sleep, rest	37.9	35.4	31.0	28.6	33.2
regular check-ups with a doc.*	58.6	59.6	55.0	48.6	55.5
drugs	27.6	27.3	34.2	26.2	28.8
regular PA	20.7	29.3	34.4	39.5	30.9
others	0.0	1.0	1.1	1.4	0.9
PA influence on medical condition					
definitely yes	55.2	46.5	66.5	76.2	61.1
cannot assess	24.1	24.2	15.9	11.2	18.9
I have not thought about it	10.3	19.2	14.4	8.5	13.1
rather no than yes	10.3	4.0	2.3	4.1	5.2
not at all	0.0	6.1	0.8	0.0	1.7
Information from physician on the importance of PA					
yes. detailed info.	48.3	30.3	28.0	26.5	33.3
yes, general info.	31.0	34.3	47.3	51.0	40.9
I have not thought about it	6.9	7.1	6.8	3.1	5.9
rather no than yes	6.9	19.2	10.0	15.3	12.8
not at all	6.9	7.1	6.8	3.1	5.9
others	0.0	2.0	1.1	1.0	1.0
Patients’ awareness of minimum requirements on PA					
yes, informed	41.4	34.3	40.6	37.8	38.5
not important	3.4	19.2	16.8	23.1	15.9
no, but interested	17.2	18.2	22.9	19.0	19.4
no information	31.0	24.2	15.1	16.3	21.7
not interested	6.9	4.0	4.2	3.7	4.7
Patients’ awareness of suitable exercises					
yes	44.8	48.5	46.3	50.3	47.5
no, but interested	10.3	19.2	20.6	25.2	18.8
not aware	31.0	31.3	30.6	22.8	28.9
not interested	13.8	1.0	2.3	1.4	4.6
others	0.0	0.0	0.2	0.3	0.1

doc. = doctor.

**Table 4 healthcare-09-01447-t004:** Leisure time activity in relation to patients’ education.

Educational Level %					
Leisure Time Activity	Elementary (*n* = 29)	Vocational (*n* = 99)	HS Graduate (*n* = 471)	University (*n* = 294)	Total (*n* = 893)
domestic chores	44.8	60.6	58.6	61.2	56.31
TV, PC	55.2	35.4	43.3	46.6	45.11
shopping	27.6	11.1	19.5	25.2	20.85
music	27.6	24.2	20.2	24.8	24.21
garden	34.5	45.5	40.8	38.1	39.70
friends	27.6	15.2	20.4	28.2	22.84
recreational PA	17.2	8.1	19.7	26.2	17.81
creative activity	0.0	7.1	5.7	6.8	4.90
reading	10.3	20.2	31.4	37.4	24.85
collecting activity	3.4	2.0	3.0	2.7	2.79
others	0.0	0.0	1.1	1.7	0.69

**Table 5 healthcare-09-01447-t005:** Frequency, duration, and type of PA performed in relation to education of NCD patients.

	Educational Level %				
Implementation of PA	Elementary (*n* = 29)	Vocational (*n* = 99)	HS Graduate (*n* = 471)	University (*n* = 294)	Total (*n* = 893)
Frequency of PA					
5 and more times/week	10.3	16.2	11.7	6.8	11.25
4 times/week	6.9	13.1	6.8	10.2	9.26
3 times/week	0.0	6.1	13.8	17.7	9.39
2 times/week	20.7	5.1	14.6	22.8	15.79
1 times/week	6.9	5.1	8.3	9.9	7.52
irregularly	44.8	47.5	36.5	27.2	39.01
no activity	10.3	7.1	8.3	5.4	7.78
Duration of PA					
less than 10 min.	10.3	14.1	10.2	7.8	10.63
10–20 min.	31.0	14.1	10.4	13.3	17.21
20–30 min.	6.9	13.1	20.8	15.0	13.95
30–45 min.	17.2	17.2	19.5	19.0	18.25
45–60 min.	17.2	17.2	17.4	29.3	20.27
60–90 min.	13.8	8.1	10.6	9.5	10.50
90–120 min.	3.4	3.0	4.0	4.1	3.65
more than 2 h	0.0	13.1	7.0	2.0	5.54
Type of PA					
domestic chores	65.5	80.8	66.5	64.6	69.35
walk	44.8	39.4	52.0	55.1	47.84
running, swimming, bike	13.8	13.1	19.3	24.5	17.68
aerobics	3.4	6.1	6.8	11.9	7.05
strengthening	13.8	5.1	10.4	16.7	11.48
sports games	0.0	4.0	4.7	4.8	3.37
others	0.0	2.0	2.3	2.4	1.68

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
