# Peer review of "Association between Educational Level and Physical Activity in Chronic Disease Patients of Eastern Slovakia"

_healthcare, 2021, doi:10.3390/healthcare9111447_

Round 1

Reviewer 1 Report

I received a manuscript for review for the title : Association Between Educational Level and Physical Activity in Chronic Disease Patients of Eastern Slovakia. The work is general and contains a few bugs. However, this topic is often discussed in the subject literature. The strength of study is the sample size which is representative.

1.    The beginning of the chapter the introduction is a duplicate of the instructions for the author
2.    Verse 44. If Slovakia is a European Union country, why separate it?
3.    The materials and methods section is well described. It contains all the required elements as well as the statistical rationale for the selection of the sample.
4.    Results, discussion, conclusions - I have no objections

Author Response

Dear reviewer,

thank you very much for your comments which meaningfully helped us to revise the manuscript. We are sincerely grateful for the valuable and constructive feedback. We have worked hard to follow your advice.

 Response to your specific comments:

  • The beginning of the chapter the introduction is a duplicate of the instructions for the author

The instructions for the author got embedded unintentionally into the article template in the introduction, our TYPO mistake, we do apologize. Additionally we have deleted the lines in word.

  • Verse 44. If Slovakia is a European Union country, why separate it?

We considered it important to inform the reader about where exactly the study was conducted.

Reviewer 2 Report

Dear Authors

I read the paper entitled "Association Between Educational Level and Physical Activity in Chronic Disease Patients of Eastern Slovakia".

The aim of the manuscript was describe the relationship between educational level and PA level in patients. This topic is well know in literature and this paper appear not add information to the current literature but shown some critical points:

  • in the abstract miss the "Method" section;
  • in the abstract were not reported any data
  • in Introduction section the line 28-38 are unecessary
  • line 45: "Adherence rates" to what?
  • Methods section appear not well organized, some information were reported both in different place of the same section. In the reviewer point of view is advisable to re-organized this section.
  • in my opinion, the section 2.1 should describe the partecipants (with inclusion and exclusion criteria) and 2.2 should describe the procedures
  • the choice of an not validated questionnaire places strong biases in the research. However, the authors claim to have drawn the questions from validated questionnaires (Behavioral Risk Factor Surveillance System and Global Physical Activity Questionnaires) please clarify.
  • In addition to previous point, it's advisable to shown the question used. 
  • please add a reference for GPAQ.
  • in results section authors should report only the data obtained from the questionnaire without making assumptions or comments that would go into the discussion section
  • in table 5 should be used  an additional parameters as VOLUME (frequency x duration) in order to provide a more solid variable of PA. In addition were not reported any statistical evaluation for the correspondence between PA and educational level, only a descriptive statistic were presented.

Author Response

Dear reviewer,

thank you very much for your comments which meaningfully helped us to revise the manuscript. We are sincerely grateful for the valuable and constructive feedback. We have worked hard to follow your advice.

We have corrected the following deficiencies identified in your assessment:

  • in the abstract miss the "Method" section
    • We additionally included the Method section in the abstract
  • in Introduction section the line 28-38 are unecessary
  • The instructions for the author got embedded unintentionally into the article template in the introduction, our TYPO mistake, we do apologize. Additionally we have deleted the lines in word.
  • line 45: "Adherence rates" to what?
  • In the process of translation, the translator added a new paragraph extending the sentence, thus causing confusion to its meaning. The term 'adherence rate' relates to the previous two sentences - NCDs and causes of death in the EU. We have additionally corrected the sentence accordingly.
  • The 'Methods' section has been modified following your suggestion. We agree, we have achieved a better overview of the section with this reorganisation.
  • We extended Methodology section by the questions used in the study.
  • We are aware that the questionnaire as such has not been validated, the questions were adopted from other validated questionnaires. We have added this comment to the 'Limitations of the study' section and also pointed out this fact in the 'Methodology' section.
  • We added a reference to the GPAQ to the 'References' list
  • In the 'Results' section, we deleted sentences that were not directly related to the results

Thank you for your recommendation to conduct further statistical analysis, however, we have already finished the project and currently we can no longer enter the process of statistical calculations.

In conclusion we only presented the variables that were found to have a significant mutual relationship. We also understand that these results do not allow us to formulate definite conclusions since this field requires more sophisticated research.

Reviewer 3 Report

The study was well done. The Paper is well written. I have minor comments:

  1. Add more limitations
  2. It could be better if the p-value is added for tables 3,4 and 5. I see that it was mentioned in the text portion of the results section. 

Author Response

Dear reviewer,

thank you very much for your comments which meaningfully helped us to revise the manuscript. We are sincerely grateful for the valuable and constructive feedback. We have worked hard to follow your advice.

Response to your specific comments:

  • we have added additional study limitations
  • The 'p-value' is not included in the table to avoid unnecessary duplication of data
  • we checked the spelling of the text and corrected spelling mistakes